# Do multidisciplinary cancer care teams suffer decision-making fatigue: an observational, longitudinal team improvement study

Tayana Soukup,[1] Tasha A K Gandamihardja,[2] Sue McInerney,[3] James S A Green,[4] Nick Sevdalis[1]

JSAG and NS contributed equally.

JSAG and NS shared the senior author position on this paper

For numbered affiliations see end of article.

**Correspondence to**
Dr Tayana Soukup;
tayana.soukup@kcl.ac.uk

## ABSTRACT

**Objective** The objective of this study was to examine effectiveness of codesigned quality-improving interventions with a multidisciplinary team (MDT) with high workload and prolonged meetings to ascertain: (1) presence and impact of decision-making (DM) fatigue on team performance in the weekly MDT meeting and (2) impact of a short meeting break as a countermeasure of DM fatigue.

**Design and interventions** This is a longitudinal multiphase study with a codesigned intervention bundle assessed within team audit and feedback cycles. The interventions comprised short meeting breaks, as well as change of room layout and appointing a meeting chair.

**Setting and participants** A breast cancer MDT with 15 members was recruited between 2013 and 2015 from a teaching hospital of the London (UK) metropolitan area.

**Measures** A validated observational tool (Metric for the Observation of Decision-making) was used by trained raters to assess quality of DM during 1335 patient reviews. The tool scores quality of information and team contributions to reviews by individual disciplines (Likert-based scores), which represent our two primary outcome measures.

**Results** Data were analysed using multivariate analysis of variance. DM fatigue was present in the MDT meetings: quality of information (M=16.36 to M=15.10) and contribution scores (M=27.67 to M=21.52) declined from first to second half of meetings at baseline. Of the improvement bundle, we found breaks reduced the effect of fatigue: following introduction of breaks (but not other interventions) information quality remained stable between first and second half of meetings (M=16.00 to M=15.94), and contributions to team DM improved overall (M=17.66 to M=19.85).

**Conclusion** Quality of cancer team DM is affected by fatigue due to sequential case review over often prolonged periods of time. This detrimental effect can be reversed by introducing a break in the middle of the meeting. The study offers a methodology based on 'team audit and feedback' principle for codesigning interventions to improve teamwork in cancer care.

### Strengths and limitations of this study

► A validated tool was used.
► Subset of cases was scored by trained evaluators in pairs blind to one another's scores.
► Main assessor was a clinician whose presence in multidisciplinary team meetings is natural.
► Observer bias and Hawthorne effect.
► Pre–post study design with no control over extraneous elements that are changing at the same time as the intervention is implemented.

## INTRODUCTION

In the UK, care planning for patients with cancer is routinely (and mandatorily) carried out by a multidisciplinary team (MDT), generally consists of histopathologists, radiologists, surgeons, specialist cancer nurses and oncologists, in typically weekly meetings (or tumour boards). Here, patients are reviewed and treatment recommendations are agreed on by the team in a sequential manner for up to a few hours at a time.[1–9] While the MDT approach to cancer care is endorsed widely,[7] evidence of its effectiveness is unclear and variable.[8–17] A pattern generally observed in MDT meetings is unequal participation to discussion and suboptimal sharing of information.[1 8–16] Evidence from studies on small groups suggests that variability in performance is attributable to human factors, such as those that are internal to teams including leadership, group composition and personality traits, as well as the external circumstances, such as increasing workload, time pressures and shifting economic landscape.[18]

Hence, one aspect of MDT meetings warrants further focus, and that is the type of fatigue that arises as a result of increasing workload. To date, evidence has documented high workloads on cancer MDTs with meetings up to 5 hours reported in the recent

Cancer Research UK report.[5] For example, in the UK, studies have reported that a breast cancer MDT reviewed between 29 and 51 patients with the meeting often running for up to 3.5 hours[1]; lung MDT between 22 and 30 patients with meetings up to 3 hours[2]; urology MDT between 19 and 51 patients with meetings up to 2 hours[3] and a colorectal MDT between 9 and 55 patients with meetings up to 1 hour and 40 min.[4] High workloads and prolonged periods of consecutive decision-making (DM) in the meetings have become a norm for many teams,[6 8] something that is likely to continue as teams are trying to maximise productivity in the face of increasing numbers of new cancer cases worldwide,[19 20] rising financial pressures[20 21] and growing staff shortages.[22]

Little is known however about the impact of such intense periods of cognitive activity on clinical performance with one study showing that the quality of endoscopy performance declines with repetitive procedures, that is, when conducted one after another for a prolonged period of time.[23] Evidence from cognitive science shows that such consecutive cognitive efforts on a task can lead to cognitive depletion, negatively affecting subsequent decisions, leading to performance decrements over time—also known as DM fatigue.[24] Consequences are many, including: rushed decisions, lack of attention to all available information and potential implications, status quo,[25 26] reduced ability to effectively evaluate choices and sustain attention, as well as easy distractibility and absent-mindedness.[27–29] Strategies, such as short breaks, consuming food, glucose and water, can help safeguard against decision fatigue,[24 30–35] something that in other industries, such as aviation, has been recognised.[34 35]

This is not the case for healthcare, however. On the one hand, WHO[36] recognises general fatigue as a leading contributor to medical error, and European Working Time Directive[37] restricts excessive night work and working hours. On the other hand, the type of fatigue that arises because of intensity and complexity of workload during working hours has not received the same level of recognition; despite healthcare being fraught with examples of intense cognitive work.[38–40] To date, the impact of DM fatigue has not been explored in healthcare settings; our objective was to examine this concept for the first time within the current study design.

One way of testing and evaluating the concept of DM fatigue with an MDT is to apply the principles of 'team audit and feedback'—a process of providing non-punitive and actionable feedback to professionals to allow them to self-assess and adjust their performance, thus stimulating desired behaviour change.[41–43] Such approach was found effective in improving practice and supporting quality improvements, and can be used to aid implementation of evidence-based interventions.[41] Within our study, this approach allowed us to elicit inputs from all team members, which we then used to codesign interventions to best meet the needs of the team in addressing DM fatigue. As a team-centred approach to intervention development, implementation and evaluation, this is, to

the best of our knowledge, yet to be applied to cancer MDTs.

## Aim and objectives

The overarching aim of our study was to identify and codesign quality-improving team interventions (in feedback sessions) and test their effectiveness (in team audits) with an MDT with high workload and prolonged meetings.

Within this overarching aim, we had two specific objectives based on the challenging circumstances the team was in with long meetings and high workload, and the scientific knowledge based on fatigue that can arise in such challenging circumstances.[23–35] It was, therefore, reasonable to explore in such concrete setting (1) the presence and impact of DM fatigue on team performance in MDT meetings and (2) the impact of a short break in MDT meetings as a countermeasure of DM fatigue.

## METHODS

### Study design

This was a longitudinal prospective observational study carried out over a 2-year period with a breast cancer MDT. Interventions were introduced within a single arm pre–post study design in order to allow us to identify and codesign interventions (in feedback session), and test whether these interventions work under difficult real-life circumstances where workload is high and meetings exceptionally long (in team audit).

### Patient and public involvement

Patients and public were not involved in the development and design of this study.

### Setting

A breast cancer MDT was recruited between 2013 and 2015 from a teaching hospital of the London (UK) metropolitan area.

### Participants

Participants were 15 members of a breast cancer team, and a total of 1335 patients with breast cancer reviewed at 30 MDT meetings. Availability sampling was used to identify the team with a criterion for the study being a cancer MDT from the UK National Health Service (NHS) that represents one of the most common types of cancer, and experiences high workload with prolonged meeting duration (>1 hour). Sample size in terms of the number of MDT meetings per study phase (n=10) was determined based on our feasibility study,[1] and a prior study of our group in urology with similar workload.[12] The study was granted Ethical Approval by the local ethics committee (JRCO REF. 157441).

### Intervention design: audit and feedback cycles

Interventions were codesigned and evaluated based on the principles of team audit and feedback.[41 42] In what follows, we outline what this process entailed.

Audit cycles focused on collecting observational data of team DM processes across three phases. In phase 1

(baseline; MDT meetings 1–10; July to November 2013), we did not introduce any interventions, but observations of care as usual. The descriptive data from this phase have been reported as a pilot study to establish feasibility of the measurement.[1] In phase 2 (MDT meetings 11–20; February to April 2014), we introduced two interventions including (1) change of the room layout from lecture theatre style to a U-shape where team members were able to face each other and (2) formal appointment of an MDT meeting chairperson. The rationale for these interventions was that the change of the room layout will be more conducive to team interactions, while appointment of the formal chair will help facilitate the overall flow of the meeting and individual patient discussions. In the final phase 3 (MDT meetings 21–30; September 2014 to March 2015), we introduced a 10 min long break for tea, coffee and snacks halfway through the MDT meetings, that is, typically at the 90 min mark, which was hypothesised to help counteract negative effects of DM fatigue.

Feedback sessions focused on identifying and codesigning interventions. The interventions were identified and chosen based on the observational data from each phase, MDT recommendations, guidelines and evidence based, as well as on team discussion and consensus within each feedback session. That is, in each feedback session, the data from previous phase were presented to the team. The data were then benchmarked against previous observational phase, guidelines, recommendations and evidence based for cancer MDTs. In the light of this information, we discussed potential evidence-based interventions that were most appropriate and acceptable to the entire team by reaching a consensus.

More specifically, the feedback sessions occurred at three time points at the end of each audit phase—in June 2014, May 2014 and June 2015. Each session was allocated a 1-hour slot as part of the MDT meeting where we (1) fed back the summary of the analysis (20 min), (2) facilitated team-based review of the findings and what they meant for the team (20 min) and (3) shortlisted evidence-based interventions the team were willing to introduce into their work in the coming study period (20 min).

The process of implementing interventions was agreed on in the feedback sessions, and it was facilitated/enabled in a collaborative manner. Specifically, following each feedback sessions, the research team produced minutes and actions that were approved and emailed to the MDT by their lead, a consultant breast surgeon (TG). The MDT was invited to comment and identify date for intervention implementation. The task of leading the introduction/implementation of the interventions was assigned to the MDT lead. Interventions were introduced and allowed a 'bed-in' period of approximately 3 months, during which no assessments were carried out to allow the team to familiarise themselves with the novel way of working. This approach was designed at the request of the MDT who needed the 'bed-in' time to ensure they did not feel they were being 'examined' by the research team at a time when they were in a state of change. The implementation process was led by the MDT, therefore.

### Materials

We used a validated quantitative observational assessment tool, namely the Metric for Observation of Decision-making, (MDT-MODe; figure 1),[10] which was tested for feasibility in our pilot study.[1] The tool has been used previously to assess various cancer MDT meetings and has shown good validity and reliability (on individual variables and composite scores).[1–4 10–14]

The MDT-MODe captures the following aspects in a meeting:

1. Quality of presented patient information, which includes six individual variables scored on a behaviourally anchored 5-point scale, namely, patients' case history, radiological images, histopathology, psychosocial issues, comorbidities and their views on treatment options. The sum of the scores for all six variables represents overall quality of presented information for a patient with the higher scores indicating better quality.
2. Quality of disciplinary contribution to patient reviews which includes six individual variables scored on a behaviourally anchored 5-point scale, representing the surgeons, oncologists, radiologists, histopathologists, Breast Cancer Nurses (BCN) and the chairperson. The sum of the scores for all six variables represents overall

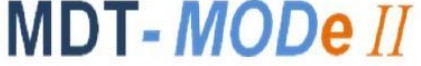
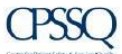

| # | Site | point | Information | | | | | | | Discussion | | | | | | | OUTCOME | |
|---|------|-------|-----|-------|------|---------|----------|----------------|----------------------------|-------|------|-------|--------|-------|---------|-----------|-------|-------------------------------|
| | | | Hx | X-ray | Path | Psy/soc/ | comorbid | Patient's view | Patient seen by a member | Chair | Surg | Phys* | Oncolo | Nurse | Radiolo | Histopath | Y/D/N | Observer aware of the decision |
| 1 | | | | | | | | | | | | | | | | | | |
| 2 | | | | | | | | | | | | | | | | | | |
| 3 | | | | | | | | | | | | | | | | | | |
| 4 | | | | | | | | | | | | | | | | | | |
| 5 | | | | | | | | | | | | | | | | | | |

**Figure 1** Metric for the Observation of Decision-making in cancer multidisciplinary team meetings (MDT-MODe).

quality of disciplinary contribution for a patient with the higher scores indicating better quality.

## Assessor training

Prior to the formal scoring during the study, the evaluator (cancer nurse specialist, SM) was trained in the use of the MDT-MODe,[10] which is a general principle for instruments assessing human factors in clinical environments.[44] Training was delivered by our team and it involved: (1) explanation of the domains, scales and their anchors, (2) background reading of peer-reviewed literature on the tool and (3) calibration of scoring against an expert evaluator (TS) via scoring a set of prerecorded MDT videos.

To ensure reliability in the use of the tool, a cross-section of the data was double-rated blindly by trained clinical (SM) and psychologist (TS) observers. To minimise Hawthorne effect, that is, teams changing their usual behaviour due to being observed, the main study evaluator was the cancer nurse specialist, the presence of whom within an MDT meeting is natural. During data collection, each evaluator was blind to the other evaluators' observations and the observer (SM) did not participate in the MDT meetings clinically. Proficiency in scoring was set as an achievement of inter-assessor reliability of 0.70 or higher between the trainee and expert assessor[44]; this was met.

## Statistical methods and variables

There were two independent variables (IVs) in the study:
► IV1 was defined as the 'study phase' with three levels (phases 1, 2 and 3) in the one-way multivariate analysis, and two levels (phases 2 and 3) in the two-way multivariate analysis.
► IV2 was defined as the 'time lapse' with two levels, namely, first and second half of the meeting.
► There were two dependent variables (DVs):
► DV1 is quality of presented patient information to the team as measured by MDT-MODe.[1 10]
► DV2 is quality of disciplinary contributions to patient review as measured by MDT-MODe.[1 10]

Three sets of analyses were conducted:
1. Intraclass correlation coefficient (ICC) analysis was used to assess reliability of evaluations in each phase. ICCs can range between 0 and 1, with higher values indicating better agreement.
2. Multivariate analysis of variance (MANOVA) was used to assess:
   A. Between-intervention differences in DM where the effect of codesigned interventions across all three phases is explored using a one-way MANOVA with post hoc tests;
   B. Within-meeting differences in DM where presence of DM fatigue and effect of a 10 min break in phases 2 and 3 is explored using two-way MANOVA with simple main effects.
3. Correlation analysis was used to ascertain presence of DM fatigue across all three phases.

All analyses were carried out using SPSS V.20.0. All pairwise comparisons are reported with Bonferroni-adjusted p values.

## RESULTS
### Meeting characteristics

The sample consisted of overall 1335 patients managed across the three study phases (see table 1). All case reviews for the duration of the study were conducted in the context of the set interventions. It is evident that the total number of patients discussed per phase steadily increased as the study progressed, which suggests increasing workload for the team over time.

### Reliability of evaluations

Agreement between evaluators was assessed on a subset of patient reviews within each phase. The selection was driven predominantly by the pragmatic considerations and the availability of the second assessor who was not a member of the participating MDT and was blinded to the patient list for the meetings and the first assessor's scores.

We used single measures interclass correlation with the two-way mixed-effects model and an absolute agreement definition. High reliability was obtained within each of the phases:
► Baseline/phase 1: information r=0.89, contribution r=0.82, n=116, 34% of the cohort.
► Phase 2: information r=0.92, contribution r=0.95, n=116, 25% of the cohort.

**Table 1** Meeting characteristics of the breast cancer team across the intervention phases

| Meeting characteristics | Phase 1 | | | | Phase 2 | | | | Phase 3 | | | |
|---|---|---|---|---|---|---|---|---|---|---|---|---|
| | Total | Mean | Min | Max | Total | Mean | Min | Max | Total | Mean | Min | Max |
| No of meetings observed* | 10 | – | – | – | 10 | – | – | – | 10 | – | – | – |
| No of patients per meeting† | 346 | 42 | 29 | 51 | 467 | 55 | 44 | 73 | 522 | 62 | 52 | 70 |
| Time per patient review (MM:SS) | – | 03:20 | 00:31 | 09:00 | – | 03:00 | 00:47 | 09:06 | – | 02:06 | 00:10 | 12:49 |
| Meeting duration (HH:MM) | – | 03:05 | 02:45 | 03:30 | – | 03:00 | 02:00 | 03:30 | – | 02:53 | 01:30 | 03:25 |

*Total N of meetings observed across all three phases=30.
†Total N of patients discusses across all three phases=1335.

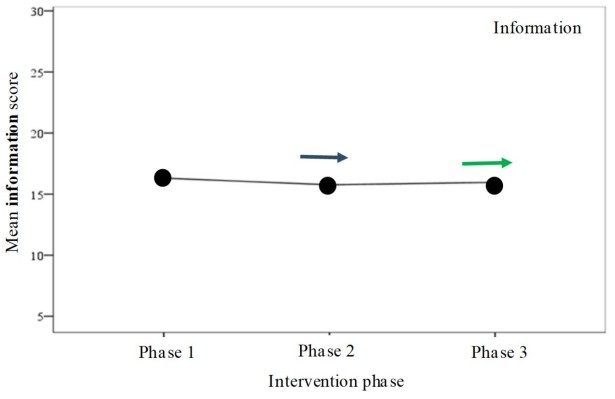

Figure 2a. Between-intervention differences on **information** scores across **all 3 phases**

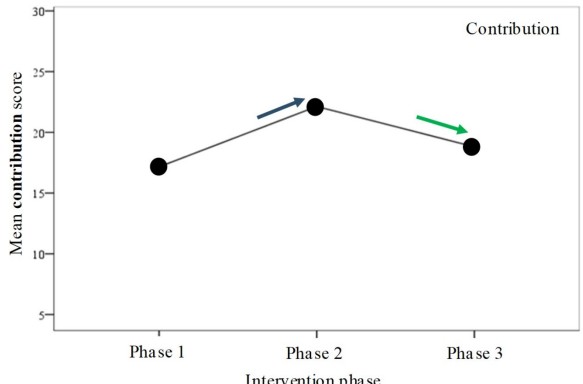

Figure 2b. Between-intervention differences on **contribution** scores across **all 3 phases**

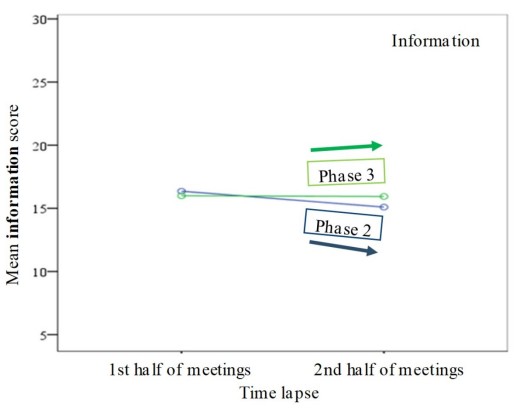

Figure 2c. Within-meeting differences on **information** scores in **phases 1 and 2**

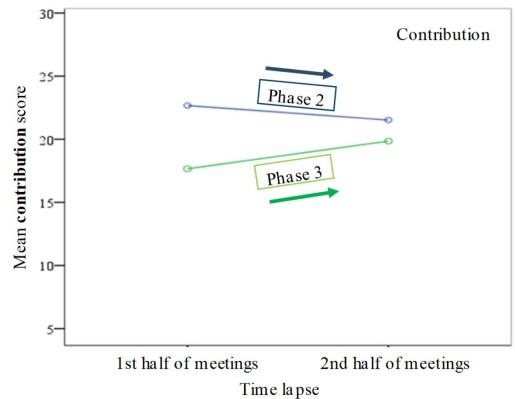

Figure 2d. Within-meeting differences on **contribution** scores in **phases 1 and 2**

**Figure 2** Mean scores for information and contribution quality across the observational phases 1, 2 and 3 (A, B), as well as across the first and second half of the meetings in phases 2 and 3 (C, D).

► Phase 3: information r=0.88, contribution r=0.79, n=131, 25% of the cohort.

## Between-intervention differences in DM across all three phases

A one-way MANOVA was run on the dataset[45] to address the overarching aim of the study, that is, to examine the effectiveness of codesigned interventions across all three study phases.

Specifically, a one-way MANOVA was run to determine the effect of codesigned interventions (IV1 with three levels: phases 1, 2 and 3) on the information (DV1) and contribution (DV2) scores of the MDT-MODe.[10] Data are expressed as mean±SD. To preserve statistical power, the Bonferroni-adjusted p level of 0.025 was used.

Information scores were similar between phase 1, 2 and 3 (16.31±3.71; 15.76±2.98 and 15.97±3.77, respectively), while the contribution scores were lower in phase 1 than 2 and 3 (17.16±3.23; 22.13±3.40; 18.81±5.50, respectively). There was a statistically significant difference between the intervention phases on the combined DVs, p<0.001.

Follow-up univariate ANOVAs showed that the information scores (figure 2A) alone did not reveal significant differences between phases (p=0.09), while the contribution scores did (p<0.025). Bonferroni post hoc tests revealed that for contribution scores (figure 2B),

phase 2 had significantly higher mean score than phases 1 (p<0.02) and 3 (p<0.02); and that phase 3 had significantly higher mean score than phase 1 (p<0.02). See figure 2A,B for a graphical representation of the results.

In sum, the findings show that the quality of information remained largely similar across phases, while the quality of contribution improved in phases 2 and 3 relative to phase 1 but with no linear improvement across phases.

## Within-meeting differences in DM in phases 2 and 3

A two-way MANOVA was run on the dataset[45] to address the two objectives in our study, that is, (1) the presence and impact of DM fatigue on team performance in MDT meetings and (2) the impact of a short break in MDT meetings as a countermeasure of DM fatigue.

Specifically, a two-way MANOVA was conducted to examine interaction effects between IV1 or a 10 min break (two levels: phase 2 meetings with no break, and phase 3 meetings with a break), and IV2 or 'time lapse' (two levels: first and second half of meetings) on the information (DV1) and contribution (DV2) scores of the MDT-MODe.[10] Data are expressed as mean±SD.

There was a significant interaction effect between 10 min break and time lapse on the information (p<0.01) and contribution scores (p<0.001). An analysis of simple main

effects for a 10 min break and time lapse was performed with significance Bonferroni-adjusted for p<0.0125. See figure 2C,D for a graphical representation of the results reported below.

Mean information scores for first and second half of meetings were 16.36±2.49 and 15.10±3.34 in phase 2, and in phase 3 they were 16.00±3.96 and 15.94±3.61, respectively. There was a significant difference in mean information scores for first versus second half of the meeting in phase 2 (p<0.001) and a non-significant difference in phase 3 when the meeting break was introduced (p=0.845). Mean information score (figure 2C) in phases 2 and 3 did not significantly differ in the first half of the meeting, 0.36 (95% CI −0.24 to 0.96) p=0.238; however, in the second half of the meeting, mean information score was significantly higher in phase 3 than phase 2, −0.84 (95% CI −1.45 to −0.24), p<0.01.

Mean contribution scores for 1first and second half of the meeting were 22.67±2.83 and 21.52±3.87 in phase 2, and in phase 3 they were 17.66±5.35 and 19.85±5.43, respectively. There was also a significant difference in mean contribution scores for first versus second half of the meeting in phase 2 (p<0.01), and also in phase 3 (p<0.001). In phase 2, mean contribution score (figure 2D) was significantly higher in the first as opposed to the second half of the meeting, 1.15 (95% CI 0.32 to 1.98), p<0.01, and in phase 3, the mean was significantly lower in the first as opposed to the second half of the meeting, −2.19 (95% CI −2.97 to −1.41), p<0.001.

In sum, quality of information and contribution was reduced in the second half of the meeting when the MDT did not have a 10 min break (phase 2). In contrast, when the MDT had a break (phase 3), the quality of information remained unchanged, while the quality of contribution improved.

### Correlation analysis: ordinal position of cases and quality of DM across study phases

A follow-up analysis was conducted on the ordinal position of cases within meetings, and information and contribution scores *to* ascertain performance decrements across all three phases, and improvements obtained in phase 3 because of a 10 min break. Ordinal position of a case within an MDT meeting is taken as an indicator of potential effects of DM fatigue: the later a case is reviewed during the MDT meeting, the more cases the team would have reviewed in a sequential manner prior to it.

Table 2 shows significant negative correlations between ordinal position of cases, and contribution and information scores in phases 1 and 2, that is, as the ordinal position of cases increases (ie, the patient is reviewed later in the meeting), the information and contribution scores decrease (ie, team interaction and clinical input measures worsen). In phase 3, however, when the short break was introduced, both coefficients are non-significant, indicating overall improvement, that is, a lack of impact of the repetitive DM process on the team interaction and clinical input indicators.

**Table 2** Pearson correlation between ordinal position of cases and the information and contribution scores

| | Information score | Contribution score | n |
|---|---|---|---|
| Ordinal position of patients in phase 1 | −0.254* | −0.160* | 346 |
| Ordinal position of patients in phase 2 | −0.206* | −0.128* | 467 |
| Ordinal position of patients in phase 3 | −0.078 | 0.072 | 522 |

n=1335 patient reviews.
*P<0.01.

Table 2 also shows that the intervention package introduced in phase 2 (change of room layout and appointing a meeting chair) did not influence the quality of DM when assessed within meetings; these effects are only detectable in the between-intervention analysis (see figure 2A,B for a graphical representation of these effects).

### Team's feedback on the conduct of the meetings

In the final feedback session (June 2015), the team recognised that the meeting break and seating rearrangement were useful and had positive impact on their working, while appointing a rotating chairperson presented with challenges and is something that would need more focus in order to ensure consistency across weekly meetings. The team reported two reasons for this, one, team friction and lack of clarity around who is chairing, and second, fatigue that the chairperson experiences by having to chair the meeting and contribute clinically to discussion ('chairing fatigue'). The team proposed that, going forward, this could be addressed by assigning the chairing role to another member of the team in the second half of the meeting.

Hence, while the fidelity of intervention delivery was good throughout—in particular for the meeting break and change of room layout which were implemented as agreed/planned in the feedback sessions, appointing a meeting chair was more challenging as it appears that although a rotating chair was appointed throughout, due to team friction, not all appointed chairs were accepted by other members of the team in the same manner.

### DISCUSSION

The overall aim of this study was to examine the effectiveness of codesigned interventions with a breast cancer team with a high workload and prolonged meeting duration, and within this, explore presence and impact of DM fatigue, and a short break as a countermeasure. Our findings were threefold. First, our study lends support for the concept of DM fatigue in MDT meetings.[23 24] In phase 2, the information and contribution quality were significantly lower in the second versus first half of the meeting. The serial positions of cases in the meetings in phases 1

and 2 were also negatively correlated with information and contribution quality, indicating performance decrements as meetings progressed. Second, our study lends support to a premise that short break in the middle of a meeting can counterbalance the effect of DM fatigue.[24] [30–35] For instance, after the break was introduced in phase 3, serial position of cases no longer showed significantly negative correlation with information and contribution quality, and the scores in the second half of the meeting no longer showed significant decrease.

Third, we found a significant increase in information and contribution quality after the introduction of codesigned interventions in phases 2 and 3 in comparison to baseline (or, phase 1). This somewhat lends support to codesigned interventions via audit and feedback.[41–43] However, a significant decrease was evident in phase 3 in comparison to phase 2, pointing to challenges at sustaining initially implemented interventions over time. In line with the final team's feedback, one explanation may be chairing fatigue and team friction, which highlights the need for continuous quality improvements and implementation science approaches to help improve our understanding of barriers and facilitators to the uptake of evidence-based interventions for cancer MDTs. It is possible that the feedback should be provided to the team at shorter intervals (after every 5th as opposed to every 10th meeting) to help reinforce the agreed change and goals. Another element that could have (also) indirectly contributed to these findings is the steady increase in workload across phases (table 2), which is known to negatively impact MDT working.[16] [17]

Nonetheless, despite the non-linear trajectory between phases 2 and 3, the improvements were made in the within-meeting performance, that is, between first and second half of the meeting in phase 3 after the 10 min break was introduced. This lends support to the concept of DM fatigue—that is, fatigue that arises because of consecutive cognitive efforts in formulating treatment recommendations, previously explored in other fields (eg, judicial DM).[24] [25] Improved quality of discussion between different disciplines is observed when break is introduced with the quality of presented patient information becoming more stable throughout the meeting. What is more, the 10 min break did not add additional time to the meeting duration (table 1), indicating that taking a break made the team more time efficient. The concept of DM fatigue has not yet been explored within cancer MDT meetings, and to our knowledge, this is the first study of its kind, with implications for the way meetings, are currently structured.

## Implications

The implications for meeting structure are far-reaching. It is the number of hours worked in a 24-hour period, and the number of consecutive hours, including the type, intensity and complexity of a task, a clinician engages in without adequate break that requires more focus and recognition. Healthcare is a highly demanding work

setting, and apart from MDT meetings, there are many examples of cognitively intense settings, including, for example, ward rounds and intensive care units.[38] [39] While the general health worker fatigue is addressed by the European Working Time Directive[37] which restricts excessive night work and working hours, the type of fatigue that arises as a result of intensity and complexity of the workload during the working hours is not adequately acknowledged or safeguarded with recommendations, such as a short break, for instance. It is understood however that the fatigue is a leading contributor to medical error and injury,[36] and that intense episodes of workload in healthcare are on the increase,[5] [19–21] as clinical teams are trying to maximise productivity in the face of severe staff shortages[22] and financial pressures.[20] [21]

## Limitations

Our findings need to be interpreted within certain limitations (some of which have been previously reported).[1]

First, participants in our study were aware that they were being observed. This was necessary due to (1) the methodological approach undertaken in our study, that is, team audit and feedback that requires the results to be fed back to the team and interventions codesigned thus making the research useful to the team, as well as (2) the ethical and regulatory constraints which meant that we had to provide full description of the study to the participants—this is due to the importance of informed consent (in line with the Good Clinical Practice), and the absence of such consent, that is, deception (eg, where MDT members are not aware that they are being observed) being regarded as high risk to participants, requiring checks and considerations by the research ethics committee that reviewed current study (where MDT members knew that they were being observed; under JRCO REF. 157441). Hence, we cannot rule out Hawthorne effect and the observer bias. While the former is a natural limitation to observational studies, we ensured that the main study evaluator was a clinician, in our case, cancer nurse specialist, the presence of whom within an MDT meeting is natural. In terms of the latter, we used a validated tool with a subset of cases scored by trained evaluators in pairs who were blind to one another's observations within each phase of the study.

Second, while this is a large-scale study for its nature (observations in real time), we acknowledge that there are cancer MDT meetings that are not as long as the ones reported here, hence the generalisability of our findings may be limited to MDTs with high workloads and prolonged meeting duration within the NHS setting. However, the global economic and healthcare landscape is rapidly changing—that is, cancer incidence[19] [20] is on the increase, as well as MDT workload,[5] [19] financial pressures[20] [21] and staff shortages.[22] The findings that we report may, therefore, become increasingly relevant to MDTs across different tumour types (and other healthcare settings) globally and could be profitably explored to determine the extent to which they apply to them.

Third, our study is of pre–post design, which can limit generalisability of our findings. This is because there is no control over other (extraneous) elements that are also changing at the same time as the intervention is implemented. While we understand that randomised controlled trials (RCTs) provide increased control of such extraneous factors allowing better precision in testing the efficacy of interventions, the aim of our study was to examine the effectiveness of interventions that were identified and codesigned with the participating team under the challenging real-world circumstances where workload and meeting duration are exceptionally high. Nonetheless, future research could adopt an RCT approach to testing the codesigned interventions identified as part of our study with multiple different MDTs to ascertain the impact of each on team functioning under ideal controlled circumstances, which in combination with our effectiveness findings with a single team under real-life circumstances would greatly enhance generalisability. However, MDTs tend to have rather different problems and priorities,[46] and so if they opt for a codesigned approach, they may end up with different interventions. Hence one would need to start off with a few smaller scale studies, such as the current one, followed by a wider consensus exercise across MDTs where a selection of team and functional improvement interventions could be identified and prioritised—these could then be designed into an RCT.

The strength of our methodological approach resides in a large sample size (n=1335), a robust methodology with validated tools and training, and an approach to improvement that is highly team centred/driven, engaging, inclusive, non-intrusive and feasible for the team (ie, does not add to their workload). Such approach has allowed us to capture complex organisational behaviour of the MDT in real time, providing good external validity, evidence of effectiveness, while identifying a set of acceptable codesigned interventions for MDTs with a high workload and increased meeting duration.

Fourth, the validated tool used in the current study (MDT-MODe) does not allow for individual person-level assessment; only disciplinary group level with the unit of analysis being a case discussion (and not an individual team member; figure 1). Such approach has advantages when evaluating a relatively small (single) team because it ensures team safety by minimising the risk of defensive routine and blaming a particular team member for performance difficulties which could, in turn, distract the team from addressing their performance problems constructively.[47] We acknowledge however that such an approach also has limitations because it does not capture (the effect of) team interaction, as well as (the effect of) individual team member's level of seniority, experience and personality, and so the effect of the physician versus the team, or style of presentation of different radiologists/histopathologists cannot be accounted for. To address these questions, a different methodological approach

may be better suited, such as conversation analysis, for instance, which allows for an in-depth analyses of team interaction on an individual person level. Also, development of tools for MDTs should take this limitation into account.

Last, while the current study is focused on DM process at the point of the MDT meeting, we have not linked these processes to clinical, patient-related outcomes. As a result, the safety implications of this analysis remain exploratory and are not yet equated to clinical outcomes.

## Further research

The objective of our study was to investigate the presence and impact of fatigue on DM processes in a team with high workload; as such, we did not address how it impacts the quality of decisions reached (eg, their clinical suitability for the patient) or patient outcomes. This is, however, an important next step that should be further explored in the light of our findings and previous research showing that DM fatigue leads to impulsive decisions, status quo and reduced ability to effectively evaluate information—these could potentially have a knock-on effect on patient outcomes.[17–22] Further research is also needed to assess the presence of DM fatigue across different cancer MDTs, particularly those with high workloads, and explore effectiveness of various evidence-based cognitive strategies.[24 30–33] Efforts should be channelled towards safeguarding optimal DM in MDT meetings, taking into account the intensity and complexity of the workload, with strategies in place as standard practice—such as, for instance, a maximum limit of cases allowed for a single meeting, mandatory short break (as practised in the aviation industry), and trained team lead/chairperson to help the team effectively navigate through workload.[6] Team-centred, codesigned approaches may prove useful in helping identify appropriate (tailored) strategies for a team, however, challenges exist at sustaining change over time; hence, a need for continuous quality improvement and implementation science approaches in the field of cancer MDTs.

## CONCLUSIONS

Previous research has shown variability in the quality of DM across cancer MDT meetings, with internal factors, such as group composition and leadership, and external circumstances, such as increased workload, time pressures and changing economic landscape held accountable. Our study demonstrates for the first time that quality of DM in cancer MDT meetings grows worse during consecutive cognitive efforts and is positively influenced with a break. Using principles of team audit and feedback to codesign team-centred interventions is a useful approach in helping initiate improvements, however, challenges exist at sustaining interventions over time. Building on our findings, further research in MDTs is needed to investigate effects of DM fatigue on the quality of decisions

reached and patient outcomes, ascertain its presence across different cancer teams, and encourage implementation of quality-improving strategies to protect optimal DM. The work could be extrapolated to other areas of clinical (and non-clinical) practice and may have implications for other areas that have equally intense periods of cognitively demanding work.

**Author affiliations**
¹Centre for Implementation Science, Health Services and Population Research Department, King's College London, London, UK
²Chelmsford Breast Unit, Broomfield Hospital, Chelmsford, Essex, UK
³Department of Cancer, Charing Cross Hospital, Imperial College Healthcare NHS Trust, London, UK
⁴Department of Urology, Whipps Cross University Hospital, Barts Health NHS Trust, London, UK

**Acknowledgements** The authors would like to thank the breast cancer MDT and their members for their time and commitment to this project.

**Contributors** In line with the guidelines by the International Committee of Medical Journal Editors, all authors for this study (ie, TS, TAKG, SM, JSAG and NS) have made substantial contributions to conception and design, or acquisition of data, or analysis and interpretation of data; have been involved in drafting the manuscript or revising it critically for important intellectual content; have given final approval of the version to be published and have agreed to be accountable for all aspects of the work in ensuring that questions related to the accuracy or integrity of any part of the work are appropriately investigated and resolved.

**Funding** Financial support for this study was provided entirely by the UK's National Institute for Health Research (NIHR) via the Imperial Patient Safety Translational Research Centre. NS and TS's research is funded by the NIHR via the 'Collaboration for Leadership in Applied Health Research and Care South London' at King's College Hospital NHS Foundation Trust, London, UK. NS is also a member of King's Improvement Science, which is part of the NIHR CLAHRC South London and comprises a specialist team of improvement scientists and senior researchers based at King's College London. Its work is funded by King's Health Partners (Guy's and St Thomas' NHS Foundation Trust, King's College Hospital NHS Foundation Trust, King's College London and South London and Maudsley NHS Foundation Trust), Guy's and St Thomas' Charity, the Maudsley Charity and the Health Foundation. The funding agreement ensured the authors' independence in designing the study, interpreting the data, writing and publishing the report.

**Disclaimer** The views expressed are those of the authors and not necessarily those of the NHS, the NIHR or the Department of Health and Social Care.

**Competing interests** NS is the Director of London Safety & Training Solutions, which provides team working, patient safety and improvement skills training and advice on a consultancy basis to hospitals and training programs in the UK and internationally. JSAG is a director of Green Cross Medical that developed MDT FIT for use by National Health Service Cancer Teams in the UK.

**Patient consent for publication** Not required.

**Ethics approval** Ethical approval for the study was given by the North West London Research Ethics Committee, and also locally by the R&D departments of the participating NHS Trusts (JRCO REF. 157441).

**Provenance and peer review** Not commissioned; externally peer reviewed.

**Data sharing statement** The anonymised dataset supporting this study is available on Zenodo, a research data repository, under the Creative Commons Attribution Non-Commercial Non-Derivative V.4.0 licence. The researchers are free to reuse and redistribute the dataset on the condition that they attribute it, that they do not use it for commercial purposes, and that they do not alter it. For any reuse or redistribution, researchers must make clear to others the licence terms of this work and cite the dataset accordingly.

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
