## [Reviewer comments · BMJ Open]

ARTICLE DETAILS

TITLE (PROVISIONAL)	Do multidisciplinary cancer care teams suffer decision-making fatigue? An observational, longitudinal team improvement study
AUTHORS	Soukup, Tayana; Gandamihardja, Tasha; McInerney, Sue; Green, James; Sevdalis, Nick

VERSION 1 - REVIEW

REVIEWER	David Hamilton Freeman Hospital, Newcastle upon Tyne
REVIEW RETURNED	16-Nov-2018

GENERAL COMMENTS	I think this is very well produced and written paper and would commend the authors. I only have a few small comments The numbering does not help the abstract objective The limitation of "RCTs are not possible" does not sit well. The intervention is NOT the presence or absence of an MDT, but MDT with changes or an MDT without. So it is possible to randomise between centres (with a redesigned MDT and one without) and within a centre (three months with it redesigned, three months without etc) Otherwise all questions which cropped up in my mind whilst reading were answered in the text - very well written
--

REVIEWER	Kenneth L Kehl Dana-Farber Cancer Institute
REVIEW RETURNED	20-Nov-2018

GENERAL COMMENTS	Overall: This is an interesting look at very granular aspects of MDT structure in cancer treatment, namely, whether decision-making fatigue occurs and might be ameliorated in the context of very long meetings. Strengths include a co-designed set of interventions, including room design, meeting leadership, and structured breaks. The principal limitation is the non-randomized design of the interventions, as the authors acknowledge. General comments: - As a US oncologist, the notion that MDT meetings might go on for several hours at a time in the UK is striking and a bit surprising. Anecdotally, our meetings rarely go on for longer than an hour.
--

	This is appropriately addressed as a generalizability concern in the discussion (page 13, last paragraph).  - Regarding the importance of the Hawthorne effect here, did MDT participants observed in each phase of intervention design know they were part of this study? Re limitations:  - Would it truly not be possible to conduct a randomized controlled trial of this question (limitations section as well as page 14, paragraph 2)? Even if MDT meetings are mandatory in the UK, is their exact structure mandatory? I don't think the fact this is not randomized means the results are not informative, so I'm not sure that particular explanation for the limitation is necessary, unless it's truly the case that the structure is also mandated. Other points:  - Page 4, last paragraph: I'm not sure the tie between MDT's and the reference regarding quality of endoscopies is clearly established in that sentence. - Page 8: was the analysis plan prespecified? - Page 10: Presentation of the results is a bit statistically dense, but the "in sum" paragraphs are a great help. For the 'within meeting' comparisons in particular, it would be nice to have the absolute information and contribution scores available in the text. For a clinical audience, perhaps the detailed F statistics could then be omitted in the main text, in favor of the P values only. (The flow might be easier to understand if the text from table 2 and the details currently in the main text were swapped). - Tables and in general: Would noted differences in information and contribution scores be considered 'clinically significant'?
--	---

REVIEWER	Pola Hahlweg University Medical Center Hamburg-Eppendorf, Department of Medical Psychology, Germany
REVIEW RETURNED	26-Nov-2018

GENERAL COMMENTS	Many thanks for the opportunity to review this study protocol. The manuscript addresses the important question of decision-making fatigue in multidisciplinary tumor conferences. However, I have several comments and suggestions to further strengthen the manuscript. General  1. Please check carefully if the use of both the abbreviations MDT and MDM is necessary. While I appreciate that they do not describe the same construct, I was wondering if the use of both might make following for the reader more complex than necessary. If you do decide to use both, please add one or two sentences to explain the distinctions. In addition, at some instances you use the term "MDT meeting". Strength and limitations  2. Limitation 2: Please revise your argument, why you used a pre-post design. In my opinion, MDMs being mandatory does not make (cluster) randomized designs impossible for your research question. Your interventions being change in seating arrangements, designation of a chair, and introduction of a break, randomization would entail randomly assigning MDMs to a group
--

of MDMs receiving those interventions (i.e. intervention group) and comparing this group to a group of MDMs that does not receive the interventions (i.e. control group).

3. Additional limitation: Please address limitations in generalizability of your results.

Introduction

4. Page 4, "Aims and objectives": Your aims and objectives seem to focus only on decisionmaking fatigue and the introduction of the short break. Please revise your aims and objectives in order to capture all the aspects your study.

Methods

5. Page 5, "Feedback sessions": spelling error "20014"

6. Page 5, "Intervention design: audit and feedback cycles":

a. How did you chose the interventions to implement? How were the interventions "codesigned"?

b. How were the interventions implemented? Did you assist implementation? Did you do process evaluation?

c. Please add information on how you collected and analyzed data throughout the team feedback process. Did you collect qualitative data (field notes, audio recordings)? How did you analyze the data?

7. Page 6, "Materials":

a. Please add one or two additional sentences to explain the MDT-MODE measure to readers that do not know it. E.g., assessment of each item for each discussed case on a 5-point Likert-scale.

1

Pola Hahlweg

University Medical Center Hamburg-Eppendorf Department of Medical Psychology

b. Please add information on how you scored the MDT-MODE within this study. Especially, since you seem to have decided to use sum scores for the two dependent variables. What was the rational to do so? Has this been done elsewhere?

Results

8. General: Did you collect any qualitative data/process evaluation data to enrich the understanding of the implementation of your interventions? I think this would be very valuable to understand your quantitative results.

9. General: Please add information on implementation outcomes such as reach and fidelity of your interventions.

10. Page 8, "Reliability of evaluations": You describe a "randomly" selected subset of patientreviews. Please describe how the subset was selected.

11. Page 8, "between-intervention differences in decision-making across all 3 phases":

a. How did you calculate the "combined DVs"? There seems to be inconsistency on whether you performed a ANOVA or MANOVA (cp. methods section and results section).

b. There is a typo in the last line of the second paragraph (Asian sign).

Discussion

12. Page 11, end of first paragraph: Please start each description of a new finding with "Firstly", "Our second finding", "thirdly" to increase readability. Putting "(our second finding)" at the end of

	the sentence makes it hard to make out where the description of the second finding begins. 13. Page 12, "Limitations": cp comment 3 14. Page 13, second paragraph: cp. comment 2 15. Page 13, "Further research", line 1: spelling error "investigateS"
--	---

VERSION 1 – AUTHOR RESPONSE

REVIEWER: 1

I think this is very well produced and written paper and would commend the authors. I only have a few small comments

- The numbering does not help the abstract objective The limitation of "RCTs are not possible" does not sit well. The intervention is NOT the presence or absence of an MDT, but MDT with changes or an MDT without. So it is possible to randomise between centres (with a redesigned MDT and one without) and within a centre (three months with it redesigned, three months without etc)

Otherwise all questions which cropped up in my mind whilst reading were answered in the text - very well written

Thank you for your positive reaction to our work. The point that you make about the RCTs is a valid one. We have now amended the reference to RCTs in the Limitations section and a detailed response on the issue of the RCTs is further elaborated on as part of our response to Reviewer 2 below. Our changes to the manuscript relating to this point appear in the Limitation's section on page 13 and under Study Design on page 4.

REVIEWER: 2

Overall:

This is an interesting look at very granular aspects of MDT structure in cancer treatment, namely, whether decision-making fatigue occurs and might be ameliorated in the context of very long meetings. Strengths include a co-designed set of interventions, including room design, meeting leadership, and structured breaks. The principal limitation is the non-randomized design of the interventions, as the authors acknowledge.

Thank you for your positive evaluation of our study. In what follows, we provide a detailed account of how we have addressed each of your comments.

General comments:

- As a US oncologist, the notion that MDT meetings might go on for several hours at a time in the UK is striking and a bit surprising. Anecdotally, our meetings rarely go on for longer than an hour. This is appropriately addressed as a generalizability concern in the discussion (page 13, last paragraph).

- Regarding the importance of the Hawthorne effect here, did MDT participants observed in each phase of intervention design know they were part of this study?

Hawthorne effect is one of the inherent limitations of observational research, and we have considered its impact on the study in quite some detail. The participants knew they were being observed as part

of a research study. The following points summarize our approach and the boundaries within which we could organize the study to minimize any bias:

(1) ethical and regulatory constraints meant that we had to provide full description of the study and its aims to the participants, as is the case for all such studies in the UK. This is due to the importance of informed consent (in line with Good Clinical Practice), and the absence of such consent i.e. deception (e.g. where MDT members are not aware that they are being observed) being regarded as high-risk to participants, requiring checks and considerations by the research ethics committee that reviewed and approved the current study (where MDT members knew that they were observed; under JRCO REF. 157441).

(2) the methodological approach we adopted was aimed to make the research useful to the team. We thus designed the study around an evidence-based 'team audit and feedback' that is effective in improving practice and supporting quality improvements with teams in complex organisational settings (explained in some detail in the Introduction, page 4). Such an approach by definition requires the findings within each phase to be fed back to the team for improved acceptability and implementation of co-designed interventions.

As the above do not mean that the Hawthorne bias was completely eliminated, we have further refined our Limitations section that covers Hawthorne effect on page 12.

Re limitations:

- Would it truly not be possible to conduct a randomized controlled trial of this question (limitations section as well as page 14, paragraph 2)? Even if MDT meetings are mandatory in the UK, is their exact structure mandatory? I don't think the fact this is not randomized means the results are not informative, so I'm not sure that particular explanation for the limitation is necessary, unless it's truly the case that the structure is also mandated.

This is a valid point, thank you for highlighting it to us. A randomised controlled trial (RCT) of the interventions introduced in the current study would indeed be possible and we would certainly encourage such approach following on from our preliminary study findings - which in addition to the evidence of effectiveness, offers a selection of co-designed interventions (i.e. change of room layout, appointing a chair, and having a break) that carry an additional value for the MDTs since they were co-designed by the core MDT members (and not imposed by us as researchers).

We have reflected on how a co-design approach could be synchronised with RCT methodology. The main issue in our view of these methods is that whilst a RCT would work best if a specific or at least a small number of improvement interventions were 'bundled' together to be evaluated for efficacy, in our experience MDTs tend to have rather different problems and priorities (Soukup et al 2018), and so if teams opt for co-design they may end up with different interventions. One way we could see the methods offering complementarity would perhaps start off with a few smaller scale studies, such as ours; followed by a wider consensus exercise across MDTs, for instance within a specific tumour type to begin with. The consensus would identify and prioritise a small number of team and functional improvement interventions, which could then be designed into a randomised trial design.

Our aim within the current study was to examine effectiveness of the co-designed quality-improving interventions in real-world circumstances (and not efficacy i.e. can interventions work under ideal circumstances), however. Within this overarching aim, the objective was to examine the presence and impact of DM fatigue that was identified in phases 1 and 2, and co-design the interventions in the feedback sessions to help mitigate negative effects given the circumstances the team was under i.e. increased workload and prolonged meeting duration. The next step in this endeavour would be a randomised controlled trial testing the efficacy of co-designed interventions following the consensus

(as described above) and in a more controlled environment in order to ascertain the level of impact each has on MDT functioning.

These methodological implications that your comment very usefully triggered are now covered in the Discussion – under Limitations on page 13, as potential further expansion of our approach. They also appear under Study Design on page 4.

Reference: Soukup, T., Lamb, B. W., Arora, S., Darzi, A., Sevdalis, N. & Green, J. S. A. (2018). Successful strategies in implementing a multidisciplinary team working in the care of patients with cancer: An overview and synthesis of the available literature. *Journal of Multidisciplinary Healthcare*. 11, 49-61.

Other points:

- Page 4, last paragraph: I'm not sure the tie between MDT's and the reference regarding quality of endoscopies is clearly established in that sentence.

Thank you for your comment. We have made an amendment to the wording in the paragraph on page 4 to increase clarity.

For more information, and to explain our thinking: we recommend the paper by Harewood et al, which has set out to explore consecutive endoscopic procedures using prospective observations. Quality indicators of colonoscopy and esophagogastroduodenoscopy were compared among procedures based on their chronological sequence. The authors found that colonoscopy completion rates declined with successive procedures; completion for 1st to 3rd procedures (90%) was significantly higher than for 4th and subsequent procedures (76%) ($P = 0.03$). Median insertion times lengthened; times for 1st to 4th procedures [8 min, interquartile range (IQR) 6-11 min] were shorter than for 5th and subsequent procedures (10 min, IQR 7-15 min) ($P = 0.06$). The authors concluded that colonoscopy cecal intubation rates decline with successive/repetitive procedures, or in other words when they are conducted one after another for a prolonged period of time. We view this as related to the repetitive cognitive task asked of busy cancer MDTs.

- Page 8: was the analysis plan prespecified?

The analysis was pre-specified, in particular ANOVA, since it was required to explain how the data will be analysed as part of the ethical approval process the study underwent prior to its conduct. The analysis had to be filled in as part of the study protocol.

- Page 10: Presentation of the results is a bit statistically dense, but the "in sum" paragraphs are a great help. For the 'within meeting' comparisons in particular, it would be nice to have the absolute information and contribution scores available in the text.

- For a clinical audience, perhaps the detailed F statistics could then be omitted in the main text, in favor of the P values only. (The flow might be easier to understand if the text from table 2 and the details currently in the main text were swapped).

Thank you for pointing this out. We have checked the results section including the table 2 and have made edits to ensure that the information and contribution scores in the text are consistently reported as mean \pm standard deviation. As suggested, we have removed F statistics in the main text in favour of P values only. We have also integrated the information from the table 2 into the text which has resulted in removing table 2 altogether to make the manuscript overall 'lighter' and easier to absorb.

- Tables and in general: Would noted differences in information and contribution scores be considered 'clinically significant'?

The differences in information and contribution are significant in terms of the quality of decision-making process at the point of the MDM. However, the safety implications of the analysis remain exploratory and are not yet equated to clinical outcomes. This is something that future studies would need to ascertain by comparing the decision-making processes at the point of the MDT meeting against the outcomes at the post-MDT stage. We have added this as a final limitation on page 14 under the Limitations section of the manuscript.

REVIEWER: 3

Comments to the Author

General

1. Please check carefully if the use of both the abbreviations MDT and MDM is necessary. While I appreciate that they do not describe the same construct, I was wondering if the use of both might make following for the reader more complex than necessary. If you do decide to use both, please add one or two sentences to explain the distinctions. In addition, at some instances you use the term “MDT meeting”.

Thank you for pointing this out to us. To enhance clarity we have removed reference to ‘MDMs’ and have changed it to ‘MDT meeting’.

Strength and limitations

2. Limitation 2: Please revise your argument, why you used a pre-post design. In my opinion, MDMs being mandatory does not make (cluster) randomized designs impossible for your research question. Your interventions being change in seating arrangements, designation of a chair, and introduction of a break, randomization would entail randomly assigning MDMs to a group of MDMs receiving those interventions (i.e. intervention group) and comparing this group to a group of MDMs that does not receive the interventions (i.e. control group).

This is a valid point, thank you for highlighting it. This was also raised by the Reviewer 2 above where we have provided a detailed response. Our changes to the manuscript relating to this point appear in the Limitations section on page 13 and under Study Design on page 4.

3. Additional limitation: Please address limitations in generalizability of your results.

Thank you for pointing this out to us. We have elaborated extensively on your point 2 above regarding the pre-post design as part of our response to Reviewer 2, and within that response we have also addressed generalizability of the results. Please see the Limitation’s section on page 13 and under Study Design on page 4.

Introduction

4. Page 4, “Aims and objectives”: Your aims and objectives seem to focus only on decision-making fatigue and the introduction of the short break. Please revise your aims and objectives in order to capture all the aspects your study.

Thank you for pointing this out to us. We have made our aim and objectives clearer. We would like to highlight that the overarching aim of our study was to identify and co-design interventions with the team as part of the feedback sessions, and then test the co-designed interventions for effectiveness as part of team audits (i.e. observations). Within this overarching aim we had two objectives which were based on the challenging circumstances of the participating team (high workload and long meetings) and the previous scientific knowledge-base on fatigue that can arise in such challenging

circumstances. It was therefore reasonable to postulate that fatigue will arise and may affect team process - focusing on decision-making fatigue and 10minute break as our objectives under the overarching aim that encompasses all co-designed interventions therefore made sense.

Methods

5. Page 5, "Feedback sessions": spelling error "20014"

Thank you, we have corrected the error.

6. Page 5, "Intervention design: audit and feedback cycles":

a. How did you chose the interventions to implement? How were the interventions "co-designed"?

Thank you so much for highlight this point to us. We have made amendments to the manuscript accordingly within the Intervention Design on page 5. More specifically, the interventions were identified and chosen based on the observational data from each phase, MDT recommendations, guidelines and evidence-base, as well as on the team discussion and consensus within each feedback session. I.e. in each feedback session, the data from previous phase was presented to the team. The data was then benchmarked against previous observational phase, guidelines, recommendations and evidence base for cancer MDTs. This was presented in a power point slide pack and prepared by the research team. In the light of this information, we discussed together in a collaborative manner the potential evidence-based interventions that were most appropriate and acceptable to the entire team by reaching a consensus, and working out the action plan in terms of how we plan to implement them and who will lead the implementation process to ensure success.

b. How were the interventions implemented? Did you assist implementation? Did you do process evaluation?

We have made slight amendments to clarify the process of implementation. The implementation process following the feedback sessions was led by the MDT clinical lead.

More specifically, the process of implementing interventions was agreed upon in the feedback sessions, and it was facilitated/enabled in a collaborative manner. Specifically, following each feedback sessions, the research team produced minutes and actions that were approved and emailed to the MDT by their lead, a Consultant Breast Surgeon (TG). The MDT was invited to comment and identify date for intervention implementation. The task of leading the introduction/implementation of the interventions was assigned to the MDT lead. Interventions were introduced and allowed a 'bed-in' period of approximately 3 months, during which no assessments were carried out to allow the team to familiarise themselves with the novel way of working. This approach was designed at the request of the MDT who needed the 'bed-in' time to ensure they did not feel they were being 'examined' by the research team at a time when they were in a state of change

We have unfortunately not conducted a detailed process evaluation – this was beyond the scope of our study and also its resources. We do however acknowledge that this is an important element that should be explored in similarly designed studies in the future.

c. Please add information on how you collected and analyzed data throughout the team feedback process. Did you collect qualitative data (field notes, audio recordings)? How did you analyze the data?

Thank you for pointing this out to us. The research team produced minutes and actions from each feedback session that were approved and emailed to the MDT by their lead, a Consultant Breast Surgeon (TG). The MDT was invited to comment and identify date for intervention implementation. This was seen as an acceptable manner to the MDT for documenting the discussion process. We

have therefore not audio or video recorded the sessions and we have therefore not subjected them to qualitative analysis – the risk of this approach would have been potential of inhibited participation in the discussion.

7. Page 6, “Materials”:

a. Please add one or two additional sentences to explain the MDT-MODE measure to readers that do not know it. E.g., assessment of each item for each discussed case on a 5-point Likert-scale.

Thank you for point this out. We have added information on the MDT-MODE under Materials page 7.

b. Please add information on how you scored the MDT-MODE within this study. Especially, since you seem to have decided to use sum scores for the two dependent variables. What was the rationale to do so? Has this been done elsewhere?

We have added information to MDT-MODE as recommended. The sum scores were used in previous research, and we have used them so that the results are easier to report and understand and also so that we can conduct multiple comparisons without losing the statistical power. For example, if we have conducted analysis on 12 individual variables (as opposed to 2 global scores), we would have ended up with 48 multiple comparisons (in contrast to 8 comparisons for within analysis) which would have made the results very dense and difficult to read and interpret (and would have also increased the chances of spurious statistical results due to numerous comparisons). Our approach to this analysis is one we have established for this psychometric instrument – e.g. Lamb BW, Green JS, Benn J, Brown KF, Vincent C, Sevdalis N. Improving decision making in multidisciplinary tumor boards: Prospective longitudinal evaluation of a multicomponent intervention for 1,421 patients. *J Am Coll Surg* 2013;217(3): 412-420.

Results

8. General: Did you collect any qualitative data/process evaluation data to enrich the understanding of the implementation of your interventions? I think this would be very valuable to understand your quantitative results.

While we understand that qualitative analyses would have enriched the understanding of implementation, we have not collected any qualitative data. The feedback sessions were minuted and were not recorded, which was in line with the team’s preferences. We do of course see the value of such further analyses – see also our earlier related response.

9. General: Please add information on implementation outcomes such as reach and fidelity of your interventions.

While we understand that measuring implementation outcomes such as reach and fidelity is important, we have not measured this formally. However, all case-reviews for the duration of the study were conducted in the context of the set interventions. Fidelity of delivery was good throughout – in particular for the meeting break and change of room layout which were implemented as intended/agreed/planned in the feedback sessions. However, appointing a meeting chair was more challenging as it appears that although a rotating chair was appointed throughout, due to team friction, not all appointed chairs were accepted by other members of the team in the same manner. We have added this information to ‘Team’s feedback’ on pages 12 and 13, as well as on page 9 under ‘Meeting characteristics’.

10. Page 8, “Reliability of evaluations”: You describe a “randomly” selected subset of patient-reviews. Please describe how the subset was selected.

The patient reviews for reliability evaluation were determined without a particular selection criterion for the cases. The second rater who was not part of the MDT chose the patients at random from the team's case list. The selection process was driven predominantly by the practical considerations by the second observer (availability and other commitments). The second observer was blinded to the patient list for the meetings and the first assessor's scores. We have clarified this further in the manuscript under the section Reliability of Evaluations.

11. Page 8, "between-intervention differences in decision-making across all 3 phases":

a. How did you calculate the "combined DVs"? There seems to be inconsistency on whether you performed a ANOVA or MANOVA (cp. methods section and results section).

Thank you for pointing this out to us. We have corrected the error and we now refer to MANOVA in both Methods and Results consistently. We do however mention 'univariate ANOVAs' in the between-intervention differences since this is commonly reported in such analyses. We have also provided more clarity on how one-way and two-way MANOVAs address the study aim and objectives.

Regarding the combined DVs on one-way MANOVA, this was determined by the statistical analysis within the SPSS and it is generated as part of the statistical output, i.e. the combined DV was determined statistically.

b. There is a typo in the last line of the second paragraph (Asian sign).

Thank you for pointing this out to us. We are not able to locate this particular typo and would be grateful if the editorial office could help us identify and correct it.

Discussion

12. Page 11, end of first paragraph: Please start each description of a new finding with "Firstly", "Our second finding", "thirdly" to increase readability. Putting "(our second finding)" at the end of the sentence makes it hard to make out where the description of the second finding begins.

Thank you, we have made amendments as suggested.

13. Page 12, "Limitations": cp comment 3

Generalisability is a valid point, thank you for highlighting it. This was also raised by the Reviewer 2 where we have provided a detailed response. Our changes to the manuscript relating to this point appear in the Limitation's section on page 13 and under Study Design on page 4.

14. Page 13, second paragraph: cp. comment 2

Thank you for pointing this out to us. We have elaborated extensively on your point 2 above regarding the pre-post design, and as part of that we have also addressed generalizability of the results. Please see the Limitation's section on page 13 and under Study Design on page 4.

15. Page 13, "Further research", line 1: spelling error "investigateS"

Thank you, this has now been corrected.

VERSION 2 – REVIEW

REVIEWER	Kenneth L Kehl Dana-Farber Cancer Institute, USA
REVIEW RETURNED	15-Feb-2019

GENERAL COMMENTS	Thank you for your thoughtful responses to our earlier comments. This is an interesting question addressed by novel methods. I have only minor further comments:  - Regarding the description of the MDT-MODE (just before Figure 1) - in item #2, the text reads "However, there was no formally appointed meeting chair in the participating team." Should this be clarified given that one of the interventions was the introduction of a meeting chair? - I'm not sure the second sentence in the "Assessor training" paragraph ("training is essential to be able to use it..") is necessary. - Results, paragraph beginning "information scores," last sentence probably should read "There was a statistically significant difference..." - Paragraph beginning "Third, our study is of pre-post..." last sentence probably should include phrase "such as the current one" rather than "such is the current one." - Paragraph beginning "follow-up univariate ANOVAs." Last sentence reads "phase 2 had significantly higher mean score than phases 1 ($p < 0.02$) and 3 ($p > 0.02$). Is the ">" a typo? Also - for those post-hoc tests: it seems surprising that phase 3 had a significantly higher mean contribution score than phase 1, even though the absolute values of the contribution scores in the preceding paragraph were quite close together with fairly wide confidence intervals and this is a univariate analysis. But perhaps I'm just missing something in the statistics?
--

REVIEWER	Pola Hahlweg Department of Medical Psychology, University Medical Center Hamburg-Eppendorf, Germany
REVIEW RETURNED	14-Feb-2019

GENERAL COMMENTS	Many thanks for the opportunity to review the revision of this manuscript. In my opinion, the changes made strengthened the manuscript significantly. However, I have a few additional comments and suggestions.  1. General: I realize throughout the manuscript that I need to pay careful attention to all the descriptions and comparisons "between", "across", and "within" phases and interventions. The reader can easily get confused. I am unsure whether it is possible to avoid this completely, but I would appreciate if you could facilitate comprehensibility. E.g. page 10, "reliability of evaluations": You state that "high reliability was obtained across all phases." Maybe "within each of the phases" would be easier to understand. 2. Page 10, "reliability of evaluations": Thank you for clarifying, how you chose the subset. However, I do not agree with this being a "randomly selected subset" and would appreciate to call it just "subset". 3. Page 11/12, "within meeting differences in decision-making in phase 2 and 3": Why do you report different comparisons for information scores and contribution scores? Please compare the last sentence of the 4th and 5th paragraph of this section. One time, you compare 1st half of phase 2 to 1st half of phase 3 (and
---

	2nd half of phase 2 to 2nd half of phase 3). The other time, you compare 1st half of phase 2 to 2nd half of phase 2 (and 1st half of phase 3 to 2nd half of phase 3). The second comparison seems to make more sense to me.
--	---

VERSION 2 – AUTHOR RESPONSE

Reviewer: 2

Thank you for your thoughtful responses to our earlier comments. This is an interesting question addressed by novel methods. I have only minor further comments:

1. Regarding the description of the MDT-MODE (just before Figure 1) - in item #2, the text reads "However, there was no formally appointed meeting chair in the participating team." Should this be clarified given that one of the interventions was the introduction of a meeting chair?

We thank the reviewer for pointing this out to us. Charring was indeed included in the analysis since it was one of the interventions. We have therefore removed this sentence from the manuscript as it is a genuine error that was overlooked during the write up of the manuscript (we worked simultaneously on a similar study).

2. I'm not sure the second sentence in the "Assessor training" paragraph ("training is essential to be able to use it...") is necessary.

We thank the reviewer for pointing this out to us. We have removed the first part of this sentence i.e. "Training is essential to be able to use it," while leaving in the statement about the importance of receiving such training in clinical environments.

3. Results, paragraph beginning "information scores," last sentence probably should read "There was a statistically significant difference..."

Thank you, we have corrected the error as suggested.

4. Paragraph beginning "Third, our study is of pre-post..." last sentence probably should include phrase "such as the current one" rather than "such is the current one."

Thank you, we have corrected the error as suggested.

5. Paragraph beginning "follow-up univariate ANOVAs." Last sentence reads "phase 2 had significantly higher mean score than phases 1 ($p < 0.02$) and 3 ($p > 0.02$). Is the ">" a typo? Also - for those post-hoc tests: it seems surprising that phase 3 had a significantly higher mean contribution score than phase 1, even though the absolute values of the contribution scores in the preceding paragraph were quite close together with fairly wide confidence intervals and this is a univariate analysis. But perhaps I'm just missing something in the statistics?

Thank you, we have corrected ">" to "<," which was indeed a typo. Regarding the second point, phase 3 indeed had a significantly higher mean contribution score than phase 1, and unfortunately, we did not achieve a linear improvement (Figure 2a and b). The improvement was achieved within meetings hence the subsequent two-way MANOVA and correlations (Figure 2c and d). The absolute values of the contribution scores are indeed close together but did not reach statistical significance at the adjusted, more stringent p value of 0.02.

Reviewer: 3

Many thanks for the opportunity to review the revision of this manuscript. In my opinion, the changes made strengthened the manuscript significantly. However, I have a few additional comments and suggestions.

1. General: I realize throughout the manuscript that I need to pay careful attention to all the descriptions and comparisons “between”, “across”, and “within” phases and interventions. The reader can easily get confused. I am unsure whether it is possible to avoid this completely, but I would appreciate if you could facilitate comprehensibility. E.g. page 10, “reliability of evaluations”: You state that “high reliability was obtained across all phases.” Maybe “within each of the phases” would be easier to understand.

Thank you, we have corrected the error as suggested.

2. Page 10, “reliability of evaluations”: Thank you for clarifying, how you chose the subset. However, I do not agree with this being a “randomly selected subset” and would appreciate to call it just “subset”.

Thank you, we have corrected the error as suggested.

3. Page 11/12, “within meeting differences in decision-making in phase 2 and 3”: Why do you report different comparisons for information scores and contribution scores? Please compare the last sentence of the 4th and 5th paragraph of this section. One time, you compare 1st half of phase 2 to 1st half of phase 3 (and 2nd half of phase 2 to 2nd half of phase 3). The other time, you compare 1st half of phase 2 to 2nd half of phase 2 (and 1st half of phase 3 to 2nd half of phase 3). The second comparison seems to make more sense to me.

Thank you for pointing this out. Different comparisons for information scores and contribution scores are an important element of the simple main effects analysis for a 10-minute break and time lapse on the two components of decision-making (i.e. the information quality, and the contribution quality) – hence the twoway MANOVA that focuses explicitly on the within-meeting differences in performance.

Regarding the sentences of the 4th and 5th paragraph, they are both, correct and important as they capture different aspects of the simple main effects comparisons that are graphically represented in Figure 2 (specifically, 2c and 2d). We highly recommend that the two sentences are interpreted in the face of Figure 2 (c and d), which appears right at the top of the page. We have made this clearer in the manuscript, and we have also updated the Figure 2 so that the individual four graphs (i.e. a, b, c, and d) flow in line with the reported results in order to enhance clarity for the reader.